# Modulation of cardiac cAMP signaling by AMPK and its adjustments in pressure overload-induced myocardial dysfunction in rat and mouse

Anne Garnier[1], Jérôme Leroy[1], Claudine Deloménie[2], Philippe Mateo[3], Benoit Viollet[4], Vladimir Veksler[1], Mathias Mericskay[1], Renée Ventura-Clapier[1☯], Jérôme Piquereau[1,5☯]*

1 UMR-S 1180, INSERM, Univ. Paris-Sud, Université Paris-Saclay, Orsay, France, 2 ACTAGen, UMS IPSIT, Univ. Paris-Sud, Université Paris Saclay, Orsay, France, 3 Physics for Medecine, Ecole Supérieure de Physique Chimie Industrielles de Paris, INSERM U1273, CNRS UMR8063, PSL University, Paris, France, 4 Université Paris Cité, Institut Cochin, CNRS, INSERM, Paris, France, 5 Laboratoire PRéTI UR 24184, Université de Poitiers, Poitiers, France

☯ These authors contributed equally to this work.
* jerome.piquereau@universite-paris-saclay.fr

**Data Availability Statement:** All relevant data are within the paper and its Supporting Information files.

## Abstract

The beta-adrenergic system is a potent stimulus for enhancing cardiac output that may become deleterious when energy metabolism is compromised as in heart failure. We thus examined whether the AMP-activated protein kinase (AMPK) that is activated in response to energy depletion may control the beta-adrenergic pathway. We studied the cardiac response to beta-adrenergic stimulation of AMPKα2-/- mice or to pharmacological AMPK activation on contractile function, calcium current, cAMP content and expression of adenylyl cyclase 5 (AC5), a rate limiting step of the beta-adrenergic pathway. In AMPKα2-/- mice the expression of AC5 (+50%), the dose response curve of left ventricular developed pressure to isoprenaline (p<0.001) or the response to forskolin, an activator of AC (+25%), were significantly increased compared to WT heart. Similarly, the response of L-type calcium current to 3-isobutyl-l-methylxanthine (IBMX), a phosphodiesterase inhibitor was significantly higher in KO (+98%, p<0.01) than WT (+57%) isolated cardiomyocytes. Conversely, pharmacological activation of AMPK by 5-aminoimidazole-4-carboxamide riboside (AICAR) induced a 45% decrease in AC5 expression (p<0.001) and a 40% decrease of cAMP content (P<0.001) as measured by fluorescence resonance energy transfer (FRET) compared to unstimulated rat cardiomyocytes. Finally, in experimental pressure overload-induced cardiac dysfunction, AMPK activation was associated with a decreased expression of AC5 that was blunted in AMPKα2-/- mice. The results show that AMPK activation down-regulates AC5 expression and blunts the beta-adrenergic cascade. This crosstalk between AMPK and beta-adrenergic pathways may participate in a compensatory energy sparing mechanism in dysfunctional myocardium.

**Funding:** The author(s) received no specific funding for this work.

**Competing interests:** The authors have declared that no competing interests exist.

## Introduction

The AMP-activated protein kinase (AMPK) is a serine/threonine kinase considered as a "cellular fuel gauge" which plays important roles in energy homeostasis regulation [1]. This kinase is a heterotrimeric protein consisting of a catalytic α subunit and two regulatory subunits, β and γ, existing under several isoforms encoded by distinct genes exhibiting differential tissue expression. AMPK is activated by increased cytosolic AMP:ATP ratio [2] through mechanisms involving allosteric activation following binding of AMP to the AMPKγ subunit, making it a better substrate for the upstream AMPK kinase liver kinase B1 (LKB1) that phosphorylates Thr172 on the AMPKα subunit [3] and worse substrate for phosphatases [4]. $Ca^{2+}$/calmodulin-dependent kinase kinase (CaMKK) pathway is an alternate activator of AMPK by phosphorylating Thr172 in response to increases in cytosolic $Ca^{2+}$ [5]. Activated AMPK acts to restore ATP levels by reducing energy consuming pathways (protein and lipid synthesis, apoptosis, cell proliferation. . .) and stimulating energy producing functions (fatty acid oxidation, glucose transport, glycolysis, mitochondrial biogenesis. . .) in order to re-establish the energy balance [1]. Newly revealed roles for AMPK further established its place at the heart of energy metabolism [6]. It is now also recognised that AMPK activity is critical for the cardiolipin composition of mitochondrial membranes, which affects the function of the mitochondrial respiratory chain [7, 8]. AMPK is mainly present in the cytoplasm but complexes containing the α2 subunit are also present in the nucleus, where they modulate gene expression [9]. Overall AMPK acts through translational (long term) and post-translational (short term) modulations impacting many cellular processes, making this kinase a major guardian of cardiac energy status.

The heart expresses both isoforms of the α-catalytic subunit α1 and α2, the latter being the predominant isoform [10]. In this organ AMPK is activated in response to physiological stimuli like exercise [11] or pathological stimuli like ischemia [12] or pressure overload [13] as well as in heart failure [14–16]. Activation of AMPK has been shown to be protective as invalidation of AMPKα2 subunit [16] exacerbates left ventricular dysfunction and hypertrophy while activation of AMPK by either metformin (an antidiabetic drug) or the pharmacological AMPK activator, 5-aminoimidazole-4-carboxamide riboside (AICAR), proved to be beneficial in the course of heart failure in dogs and mice [14, 15]. More recently, studies using mouse models of AMPK deficiency reinforced the idea that AMPK is essential for cardiac functions and is cardioprotective when the heart is under stress [17]. While many pathways have been explored to understand the benefits of AMPK functions in the heart, we are convinced that they are not understood in all their complexity and that some targets of AMPK have yet to be identified. In a large cardiac genome wide screen of AMPKα2-/-mice (http://www.ncbi.nlm.nih.gov/geo/query/acc.cgi?acc=GSE18843) we have identified the adenylyl cyclase (AC) type 5 (but not type 6) as a putative new target of AMPK in the heart. We thus thought to investigate this interaction which could be of importance in a pathological context where the heart is under adrenergic hyperstimulation.

Cyclic AMP, the product of AC is a crucial intracellular signal that regulates heart rate and contractility. It is generated by nine membrane-bound or one soluble adenylyl cyclase. AC5 and 6 are the main cardiac isoforms [18–20] that are rate-limiting steps in the β-receptor system, with AC5 being the dominant isoform in the adult heart. ACs play a key role in determining the cardiac response to a variety of stimuli, in particular, the sympathetic stimulation [21].

β-adrenergic stimulation initiates the most potent stimulus for enhancing cardiac output, both acutely by inducing chronotropic and inotropic effects, and chronically by regulating gene transcription. It plays a major role in heart failure (HF) because there is an over-activation of the sympathetic system in patients. Clinically, β-blockers have been successfully used

for decades in HF patients and their efficiency could be especially explained by a block of the detrimental consequences of sustained β-receptor stimulation and/or a resensitization of the cardiac β-adrenergic cascade [22]. Moreover, one characteristic of the failing heart is a disequilibrium of the energy balance with energy starvation [23], high-energy phosphate depletion [24], altered energy fluxes [25] and decreased mitochondrial biogenesis [26]. By reducing heart rate, regulating calcium homeostasis, and improving diastolic filling and blood flow, β-blockers exert beneficial effects on the energetic balance.

Considering both dysregulations of the β-adrenergic pathway and of energy metabolism in HF, the understanding of the role of AMPK in cAMP signaling evoked by the fact that AC5 is a potential target of this kinase is crucial. The aim of the present study was thus to decipher the possible interaction between AMPK activation and the β-adrenergic pathway in the healthy and dysfunctional heart.

## Methods

### Animal models

Generation of AMPKα2-/- mice has been described elsewhere [27]. Mice were backcrossed on C57B6 background (≥8 times). Seven-month-old male AMPKα2-/- and control littermate mice (WT) were used.

Pressure overload was induced in seven-week-old mice to estimate the ability of AMPK deficient mouse hearts to adapt to pathological stress. Anesthesia was induced by intraperitoneal injection of Ketamine (50mg.kg$^{-1}$) and Xylazine (8mg.kg$^{-1}$). Transverse aortic constriction was induced by placing a silk suture around the aorta after thoracic incision (TAC group; WT-TAC, n = 8; KO-TAC, n = 6). Age-matched controls underwent the same procedure without placement of suture (sham group) (WT-SH, n = 6; KO-SH, n = 6). Mice were studied ten weeks after surgery.

Heart failure was induced by aortic constriction in weaned male rats by placing a stainless steel hemoclip of 0.6mm ID on the ascending aorta via a thoracic incision, as previously described [21]. After 5 months, the surviving HF animals showed increased heart and lung weights, ascites, pleural effusion, and edema indicating severe heart failure [28].

Animals were housed under temperature-controlled conditions (21˚C) and had free access to water and to a standard mouse chow. All animal experimental procedures were approved by animal ethics committee of Paris-Sud University, authorized by French government (authorization number: B9201901) and complied with directive 2010/63/EU of the European Parliament on the protection of animals used for scientific purposes.

### Echocardiography

Transthoracic echocardiography was performed in mice using a 12 MHz transducer (Vivid 7, General Electric Healthcare) under 2.5% isoflurane gas anesthesia. Two-dimensional-guided (2D) M-mode echocardiography was used to determine wall thickness and left ventricular chamber volume at systole and diastole and contractile parameters, such as ejection fraction (EF).

### Quantitative real-time PCR

Total RNA was isolated from frozen cardiac cells or tissue samples using the Trizol reagent technique according to the manufacturer's instructions (Invitrogen, Cergy Pontoise, France). Single-stranded cDNA was prepared from 4–5 μg of total RNA, using an oligo-dT primer and the Superscript II enzyme (Invitrogen), according to the manufacturer's instructions. Real-

time PCR was performed using the SYBR®Green technology on a LightCycler rapid thermal cycler (Roche Diagnostics) as described [26, 29]. Forward and reverse primers were designed in a different exon of the target gene sequence, eliminating the possibility of amplifying genomic DNA using Primer 3 Plus tool (AC5 forward: 5'-GACAACGCTGACCTTCTGGT-3', reverse: 5'-CCTGCAGTTTCCAGAGGAAG-3'; AC6 forward: 5'-CCCCGTGTTCTTC GTCTACA-3', reverse: 5'-GGAAGAGCACCACGTTAGCA-3'; reference genes TATA-box binding protein (TBP) forward: 5'-GGCCTCTCAGAAGCATCACTA-3', reverse: 5'-GC CAAGCCCTGAGCATAA-3'; polymerase II subunit a (Polr2a) forward: 5'-CAA CAT GCT GAC AGA TAT GAC C-3', reverse: 5'-TGA TGA TCT TCT TCT TGT TGT CTG-3'; and 14-3-3 protein zeta/delta (Ywhaz) forward: 5'-AGA CGG AAG GTG CTG AGA AA-3', reverse: 5'-GAA GCA TTG GGG ATC AAG AA-3'). For each set of primers, a basic local alignment search tool (BLAST) search revealed that sequence homology was obtained only for the target gene. Prior optimization was conducted for each set of primers, which consisted of determining optimal primer and $MgCl_2$ concentrations, the template concentration and verifying the efficiency of the amplification. To confirm the specificity of the amplification, the PCR product was subjected to a melting curve analysis and agarose gel electrophoresis. PCR amplification was performed in duplicate in a total reaction volume of 15μl. The reaction mixture consisted of 5μl diluted template, 3μl FastStart DNA Master$^{Plus}$ SYBR Green I kit (X5), 1.5mM $MgCl_2$ and 0.5μM forward and reverse primers. After an 8-min activation of Taq polymerase, amplification was allowed to proceed for 30–40 cycles, each consisting of denaturation at 95°C for 5s, annealing at 55–60°C for 5s and extension at 72°C for 5–8s, depending on the target gene. Quantification results for each gene were normalized to TATA-box binding protein gene expression and to Polr2a and Ywhaz for mouse TAC experiments.

## Perfused heart

Mouse were anesthetised by intraperitoneal injection of pentobarbital (100mg.kg$^{-1}$). The heart was quickly removed, cannulated and perfused by the Langendorff method with a latex water-filled balloon inserted into the left ventricular chamber and connected to a pressure transducer (Statham gauge Ohmeda, Holland) as described previously [30]. The ventricular systolic pressure was measured on-line, and computed by Emka data analyser (Emka Technologies, Paris, France). To estimate contractility on basal conditions, the latex balloon inserted inside the ventricle was inflated progressively to maximal isovolumic condition of work. To estimate AC and β-adrenergic response, 100μM forskolin and increasing concentrations of isoproterenol from 1nM to 1μM were infused in the heart.

## Cardiomyocyte isolation and culture

Individual adult ventricular myocytes (AVMs) were obtained by retrograde perfusion as previously described [31]. Male Wistar rats were anesthetized by intraperitoneal injection of sodium pentobarbital (100mg.kg$^{-1}$), and hearts were rapidly excised. The ionic composition of the $Ca^{2+}$-free Ringer solution was (in mM): NaCl 117, KCl 57, $NaHCO_3$ 4.4, $KH_2PO_4$ 1.5, $MgCl_2$ 1.7, D-glucose 11.7, sodium phosphocreatine 10, taurine 20, and HEPES 21, adjusted to pH 7.1 with NaOH at room temperature. For enzymatic dissociation, 1mg/ml collagenase A (Boehringer Mannhein, Mannhein, Germany) and 300mM EGTA were added to the $Ca^{2+}$-free Ringer solution, so that the free $Ca^{2+}$ concentration was adjusted to 20mM. The hearts were retrogradly perfused at a constant flow of 6ml/min and at 37°C by $Ca^{2+}$-free solution during 5min followed by 1h of perfusion at 4ml/min with the same solution containing collagenase. The ventricles were then separated from atria, finely chopped and gently agitated to dissociate individual cells. The resulting cell suspension was altered on a gaze and the cells were allowed to

settle down. The supernatant was discarded and cells were resuspended four more times in $Ca^{2+}$-free solution containing a progressively increasing calcium concentration. The cells were maintained at 37˚C until use.

Freshly isolated cells were suspended in minimal essential medium (MEM: M 4780; Sigma, St Louis, MO USA) containing 1.2mM $Ca^{2+}$, 2.5% fetal bovine serum (FBS, Invitrogen, Cergy-Pontoise, France), 1% penicillin-streptomycin and 2% HEPES (pH 7.6) and plated on 35mm laminin coated culture dishes (10μg/mL, 2h). After 2h the medium was replaced by FBS-free MEM.

For patch-clamp experiments, individual mouse AVMs were obtained by retrograde perfusion. Aorta were cannulated and the heart was perfused using retrograde Langendorff perfusion at 37˚C with an oxygenated $Ca^{2+}$-free Tyrode solution and digested with Liberase Blendzyme 3 (Roche) for 8min. Cells were suspended in Tyrode's solution supplemented with 5mg/ml BSA with progressive increase in calcium concentration to 1mM $Ca^{2+}$.

## Electrophysiological experiments and live cell imaging

The whole cell patch-clamp technique was used to record the long-lasting calcium current ($I_{Ca,L}$). Patch electrodes resistance was between 0.5–1MΩ when filled with internal solution containing (in mM): CsCl 118, EGTA 5, $MgCl_2$ 4, sodium phosphocreatine 5, $Na_2ATP$ 3.1, $Na_2GTP$ 0.42, $CaCl_2$ 0.062 (pCa 8.5), HEPES 10, adjusted to pH 7.3. Extracellular $Cs^+$-Ringer solution contained (in mM): NaCl 107.1, CsCl 20, $NaHCO_3$ 4, $NaH_2PO_4$ 0.8, D-glucose 5, sodium pyruvate 5, $CaCl_2$ 1.8, $MgCl_2$ 1.8 and HEPES 10, adjusted to pH 7.4. The cells were depolarized every 8s from –50 to 0mV during 400ms. Potassium currents were blocked by replacing all K+ ions with external and internal $Cs^+$. Voltage-clamp protocols were generated by a challenger/09-VM programmable function generator (Kinetic Software, Atlanta, GA, USA). The cells were voltage-clamped using a patch-clamp amplifier (model RK-400; Bio-Logic, Claix, France). Currents were analog filtered at 3KHz and digitally sampled at 10KHz using a 12-bit analogue-to-digital converter (DT2827; Data translation, Marlboro, MA, USA) connected to a compatible PC (386/33 Systempro; Compaq Computer Corp., Houston, TX, USA). The non selective phosphodiesterase inhibitor 3-isobutyl-l-methylxanthine (IBMX) was applied for ≈5min and washed for 5 to 10min until recovery of a new steady state. The maximal amplitude of $I_{Ca,L}$ was measured as the difference between the peak inward current and the current at the end of the 400 ms duration pulse. Currents were not compensated for capacitance and leak currents. Mean capacitance was 203±17 for WT and 203±10pF for AMPKα2-/- AVMs (n = 15 for each).

## Live cell imaging

FRET measurements were performed on myocytes infected with an adenovirus encoding Epac2-camps (MOI = 1000pfu/cell) for 48 hours as previously described [31]. Cells were maintained in a $K^+$-Ringer solution containing (in mM): NaCl 121.6, KCl 5.4, $MgCl_2$ 1.8, $CaCl_2$ 1.8, $NaHCO_3$ 4, $NaH_2PO_4$ 0.8, D-glucose 5, sodium pyruvate 5, HEPES 10, pH 7.4 with or without AICAR (1 mM) for 48 hours. Images were captured every 5s using the 40x oil immersion objective of a Nikon TE 300 inverted microscope connected to a software-controlled (Metafluor, Molecular Devices, Sunnyvale, CA, USA) cooled charge coupled (CCD) camera (Sensicam PE, PCO, Kelheim, Germany). CFP was excited during 300ms by a Xenon lamp (100W, Nikon, Champigny-sur-Marne, France) using a 440/20BP filter and a 455LP dichroic mirror. Dual emission imaging of CFP and YFP was performed using an Optosplit II emission splitter (Cairn Research, Faversham, UK) equipped with a 495LP dichroic mirror and BP filters 470/30 and 535/30, respectively. Average fluorescence intensity was measured in a region of

interest comprising the entire cell or a significant part of the cell. Background was subtracted and YFP intensity was corrected for CFP spillover into the 535nm channel before calculating the CFP/YFP ratio. Ratio images were obtained with ImageJ software (National Institutes of Health). Data are represented as mean±SEM.

## Western blotting

Immunoblot protein levels of phosphorylated αAMPK Thr 172, total αAMPK, phosphorylated acetyl CoA carboxylase (ACC) Ser79 and total ACC were determined by using anti-panα and anti-phosphorylated αAMPK Thr 172 antibodies (#2532 dilution 1/1500 and #2531 dilution 1/1500, respectively) and ACC and anti-phosphorylated ACC Ser79 antibodies (#3662 dilution 1/1500, and #3661 dilution 1/5000, respectively) obtained from Cell Signaling Technology (Ozyme, France). Briefly, protein extracts were loaded onto SDS-polyacrylamide gels and separated for 120min at 120V. After electrophoresis, the proteins were transferred to Hybond nitrocellulose membranes (Amersham) using a Bio-Rad blot system for 90min at 150V. Membranes were incubated overnight with the primary antibody at 4°C and with a rabbit anti-IgG HRP antibody (dilution 1/1500; Calbiochem-VWR ref 401315) for one hour. After incubation, membranes were revealed with enhanced chemiluminescent substrate (PIERCE dura, Fischer, France). Loading controls are provided to show that the amount of loaded proteins was equal for all groups. Light emission was detected by autoradiography and quantified using an image-analysis system (Chemidoc XRS, Biorad).

## Statistical analysis

Statistics were performed using Statistica software. Statistical significance was evaluated using Student's unpaired t test or after one-way ANOVA followed by Newman-Keuls test where appropriate. Curve fittings of isoprenaline dose/response were compared with Fisher test. The difference was considered statistically significant when $p < 0.05$.

## Results

### Identification of the genes modulated in *AMPKα2$^{-/-}$* mice

In a preliminary study, using mouse pan-genomic DNA microarrays (http://www.ncbi.nlm.nih.gov/geo/query/acc.cgi?acc=GSE18843), we identified AC5, as one significantly up-regulated gene (x1.5) in AMPKα2-/- hearts. Upregulation of the AC5 gene expression in KO mice was confirmed by real time RT-PCR (Fig 1A, 144%, $p < 0.001$). Conversely, AC6 the second adenylyl cyclase expressed in the heart was not altered (Fig 1B, 107%, ns). This suggests a direct or indirect link between AMPK and the β-adrenergic signaling. We thus studied the cardiac response of AMPKα2-/- mice to activation of this pathway. As we could not qualify commercially available antibodies as specific for AC5 because they cross-react with AC6, we next investigated the functional consequences of AC5 modulation.

### *AMPKα2$^{-/-}$* cardiomyocytes have an enhanced response of $I_{Ca,L}$ to the phosphodiesterase (PDE) inhibitor 3-isobutyl-l-methylxanthine (IBMX)

One of the main targets of AC-dependent cAMP increase is protein kinase A (PKA). PKA phosphorylates several proteins that are essential for cardiac function, among which are the L-type calcium channels. Thus the measure of $I_{Ca,L}$ is a surrogate for the measure of subsarcolemmal cAMP levels [30] and a proof for the physiological relevance of altered AC activity. To assess whether up-regulation of AC5 in the absence of AMPKα2 was of physiological significance, we investigated $I_{Ca,L}$, one of the main target of AC-induced cAMP increase in the

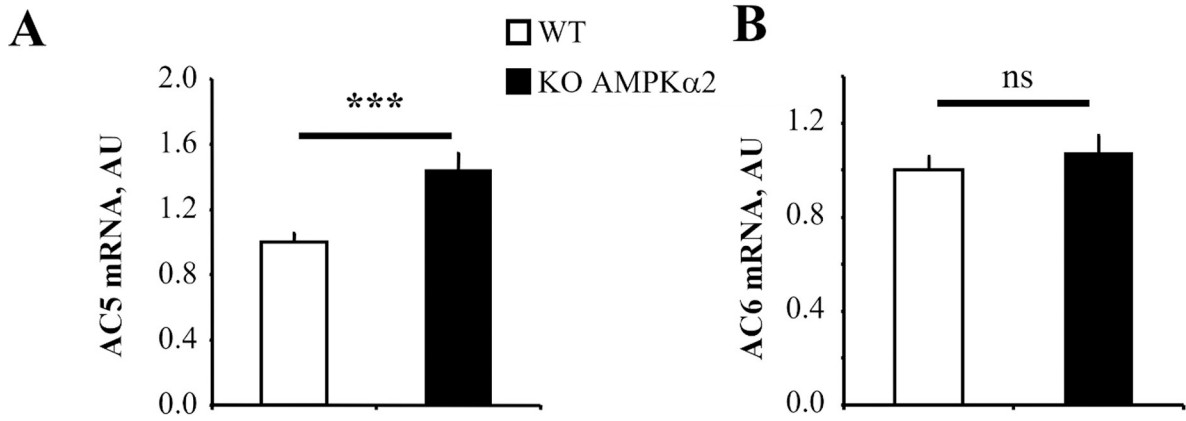

**Fig 1. Expression of AC5 and AC6 in WT and AMPKα2-/- mice.** Cardiac mRNA content of AC5 (left panel) but not AC6 (right panel) normalized to TBP is increased in AMPKα2-/- (n = 10) compared to WT (n = 11) mice. Values are means±SEM. ***p<0.001 vs. WT.

mouse adult ventricular myocytes (AVMs) by conventional whole-cell patch-clamp technique (Fig 2). While basal calcium current was not significantly different between WT and KO mice (Fig 2C), PDE inhibition by the broad-spectrum inhibitor IBMX, had a more prominent effect on $I_{Ca,L}$ recorded from AMPKα2-/- than WT cardiomyocytes (+72%, p<0.01) (Fig 2D), evidencing an increased basal activity of adenylyl-cyclase.

### $AMPK\alpha2^{-/-}$ mice have an enhanced contractile response to beta-adrenergic stimulation

Contractile and calcium homeostasis proteins are highly sensitive to cAMP dependent phosphorylation. We thus examined whether the contractile properties of the heart were more sensitive to β-adrenergic stimulation in the absence of the α2 subunit of AMPK. The contractile response of AMPKα2-/- hearts to adrenergic stimulation was studied in isolated Langendorff perfused hearts from WT and KO mice. As shown in Fig 3A, while no difference was observed on basal developed pressure, the AC activator forskolin (FSK, $10^{-4}$M) increased developed pressure more in KO than in WT mice (+25%, p<0.01). Moreover, the dose response curves to isoprenaline (Fig 3B) also showed a greater isoprenaline-induced increase in developed pressure in KO than in WT mice (p<0.001). These results clearly show a functional relevance for the upregulation of adenylate cyclase activity in absence of AMPKα2.

### Activation of AMPK in rat AVMs decreases AC5 gene expression

The next series of experiments were performed to establish whether in turn AMPK activation was able to modulate AC5 expression and cAMP content. Treatment of isolated rat AVMs for 48 hours with the AMPK activator AICAR increased AMPK phosphorylation with no change in total AMPKα protein content (Fig 4A); this phosphorylation did confirm AMPK activation by AICAR treatment. AICAR significantly reduced AC5 expression by 45% (p<0.001, Fig 4B).

### Activation of AMPK in rat AVMs decreases cAMP content

To compare basal cAMP levels with or without AMPK activation with AICAR (1mM) for 48 hours, Epac2-camps was expressed in rat AVMs using recombinant adenovirus. This sensor can detect rapid changes in cAMP with high sensitivity (EC$_{50}$≈1μM). As shown in Fig 5, application of IBMX (100μM) produced a higher increase in cAMP in control cells (Fig 5A) compared to AICAR treated cells (Fig 5B) as monitored by FRET changes between CFP and YFP.

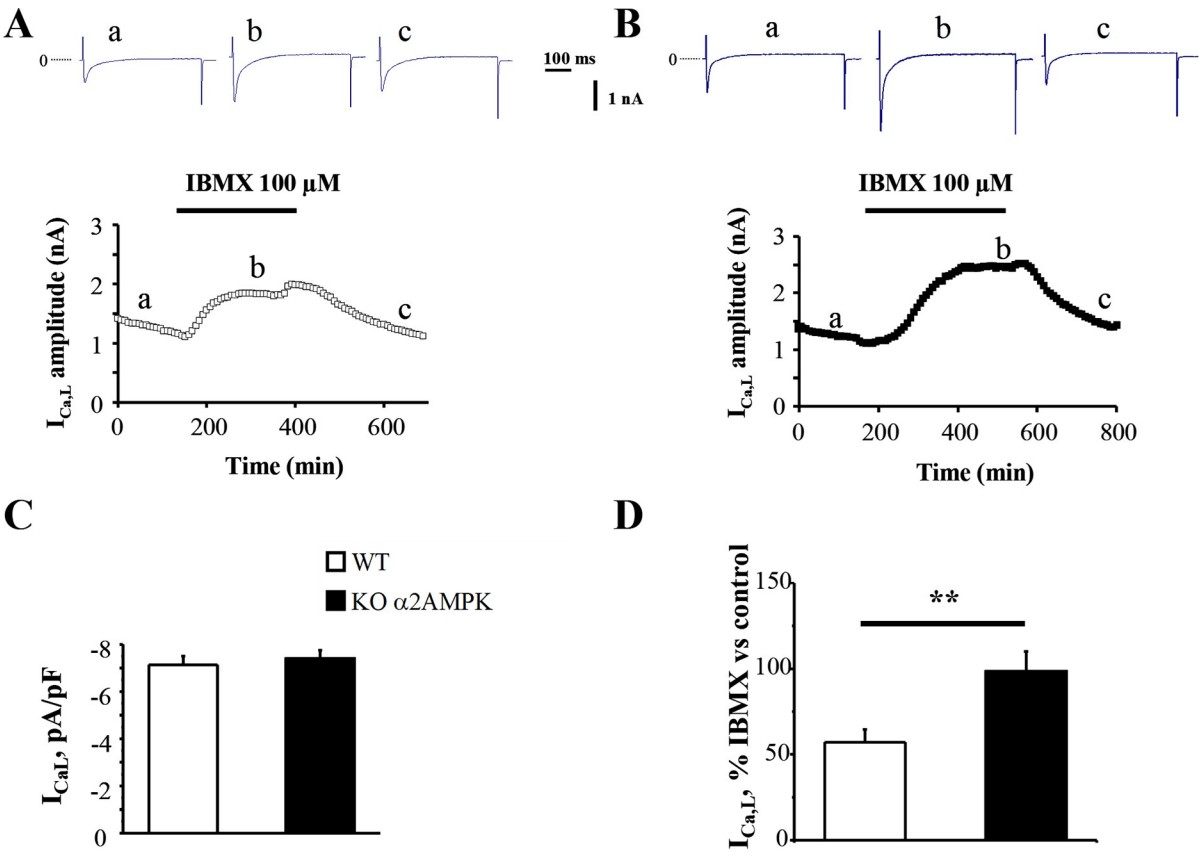

**Fig 2. Response of $I_{Ca,L}$ to IBMX in WT and AMPKα2-/- mice AVMs.** Myocytes from WT (**A**) and AMPKα2-/- mice (**B**) were first exposed to the control external solution (a) and then to 100μM of the non-specific PDE inhibitor IBMX (b) before recovery (c); solid lines indicate the period of drug perfusion. Each symbol corresponds to a measure of $I_{Ca,L}$ at 0 mV obtained every 8 s (holding potential of -50 mV). The individual current traces shown on top of the graphs were recorded at the times indicated by the corresponding letters in the bottom graphs. **C,** the mean basal $I_{Ca,L}$ was not different between groups. **D**, IBMX induced a higher increase in $I_{Ca,L}$ in KO than WT mice; (n = 15 AVMs for WT and AMPKα2-/- isolated from 5 WT and 3 KO mice). **p<0.01 vs. WT.

A 40% lower IBMX-induced change in CFP/YFP ratio was observed in AICAR-treated AVMs versus control (p<0.001) (Fig 5C), showing that AMPK activation decrease cAMP turnover in living cells.

## AC5, AMPK and cardiac stress

We considered investigating whether AMPK is activated and AC5 is down-regulated in an experimental model of systolic dysfunction (SD) induced by aortic constriction in rats. The phosphorylated form of AMPKα (Fig 6A and 6B) was significantly increased by +64% (p<0.05) in SD compared to sham, in parallel with the increase in total AMPKα (+63%, p<0.01) while the ratio of p-AMPKα/AMPKαtotal remained constant. AMPK activation was confirmed by the increased phosphorylation of its target acetyl-CoA carboxylase (ACC; Fig 6C, +170%, p<0.001). Associated with AMPK activation, AC5 expression exhibited a 40% drop (p<0.01, Fig 6D) in SD compared to sham rats.

To confirm that AC5 downregulation was linked to AMPKα2 activation, pressure overload was induced by transverse aortic constriction (TAC) in WT and AMPKα2-/- mice. Ten weeks after TAC, AMPKα2-/- mice developed similar hypertrophy (Fig 7A) but exhibited worsening of cardiac function (Table 1) as shown by greater reduction in ejection fraction as compared

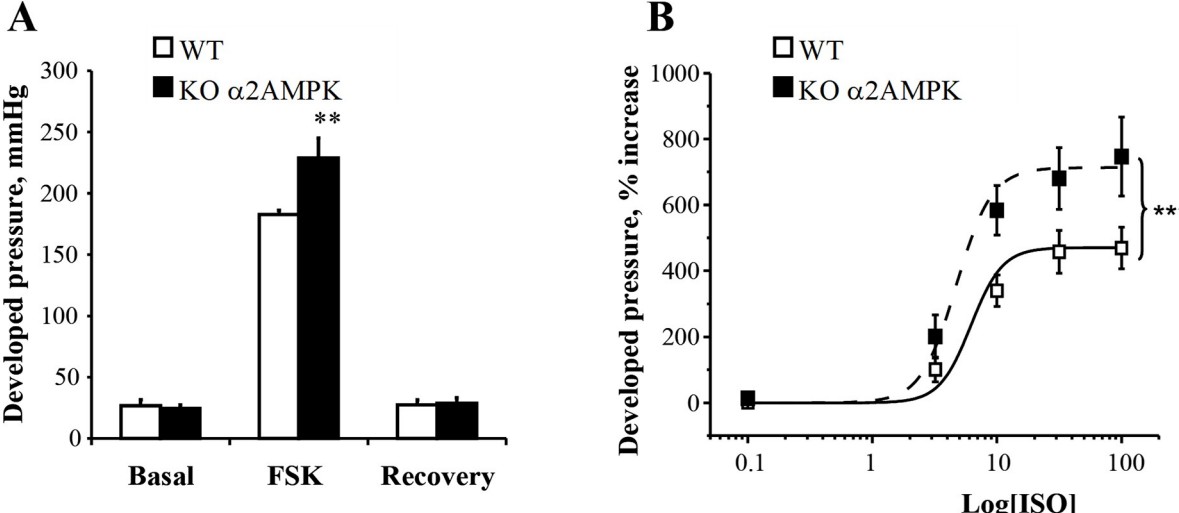

**Fig 3. Response of developed pressure to forskolin and isoprenaline of isolated perfused hearts. A,** developed pressure (mmHg) of isolated perfused hearts before, during and after application of forskolin (FSK, $10^{-4}$M). **B,** developed pressure in response to increasing doses of isoprenaline ranging from $10^{-10}$ to $10^{-7}$M. Mean steady state values obtained 10 minutes after the change in isoprenaline concentration (□: WT, n = 6; ■: AMPK$\alpha$2-/-, n = 5). The individual curves were fitted to the Hill equation and compared using Fisher test. ***$p < 0.001$ vs. WT.

with WT mice (Fig 7B). As expected, AMPK was activated by pressure overload in WT (Fig 7C) as shown by a 58% increase in p-AMPK$\alpha$ ($P < 0.01$) and a 33% ($P < 0.01$) increase in ACC phosphorylation. As in the rat, the total AMPK$\alpha$ content was increased in WT mice while the ratio of p-AMPK$\alpha$/AMPK$\alpha$total remained constant (0.90±0.16 in WT-SH versus 0.69±0.05 in WT-TAC, ns). Conversely, in KO mice, total and phosphorylated AMPK$\alpha$ was largely decreased and was not activated by TAC. The remaining AMPK$\alpha$ band reflects the AMPK$\alpha$1 isoform [24] that was not activated by TAC as shown by the constant level of phosphorylated AMPK and ACC. Incidentally, the low level of AMPK$\alpha$ content in AMPK$\alpha$2-/- group confirmed again that AMPK$\alpha$2 is largely predominant in the heart.

While AC5 expression exhibited a 23% drop ($p < 0.05$) in WT-TAC, it was not significantly altered in KO-TAC mice (Fig 7D). AC6 expression was significantly down-regulated by 33%

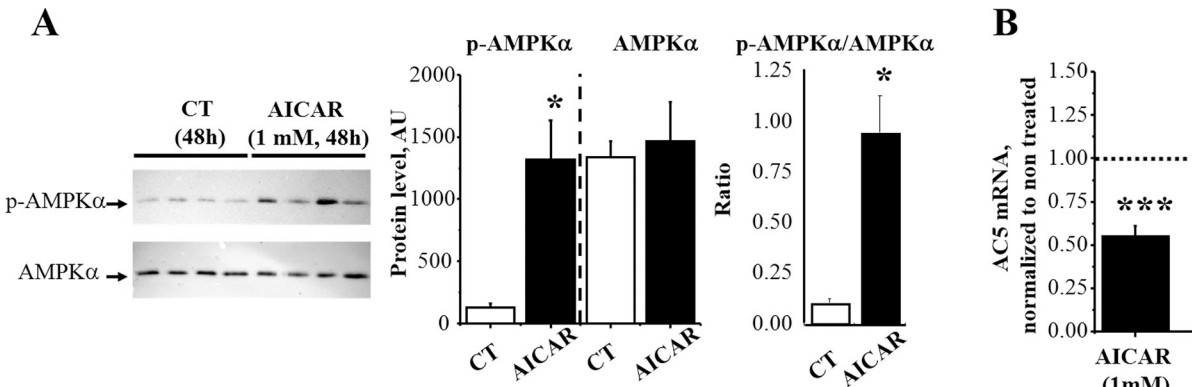

**Fig 4. AMPK phosphorylation and AC5 expression in rat AVMs in response to AMPK activation.** Rat AVMs were incubated for 48h with 1mM AICAR. **A,** Western blot analysis with phosphorylated-$\alpha$AMPK and pan-$\alpha$AMPK antibodies evidenced an increase in AMPK phosphorylation in response to AICAR. **B,** Adenylyl cyclase 5 (AC5) gene expression was measured by real-time RT-PCR and normalized to TBP. Values are means±SEM of five independent cultures per experimental condition. *$p < 0.05$ and ***$p < 0.001$ for differences from control values.

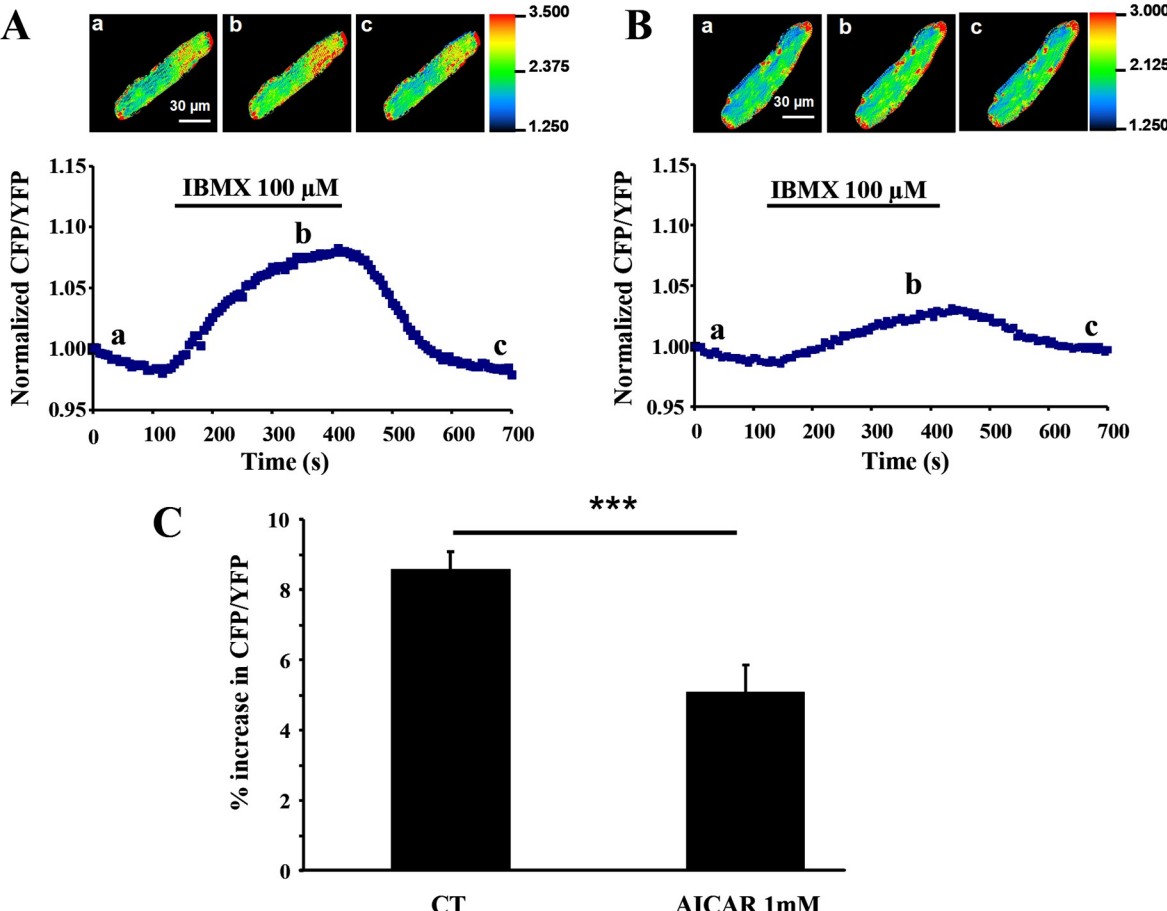

**Fig 5. Real-time measurements of cAMP in response to IBMX in control and AICAR-treated rat AVMs.** Non-treated (**A**) or AICAR-treated (**B**) rat AVMs were infected with cAMP sensor Epac2-camps adenovirus for 48h to estimate cAMP content. Following CFP excitation at 440±20nm, cAMP binding to the sensor induces a transient decrease in fluorescence resonance energy transfer (FRET) between CFP and YFP plotted as the ratio of the corrected CFP/YFP fluorescence (see Materials and Methods). Acquisitions were performed every 5 seconds. Pseudocolor images reflecting the CFP/YFP ratio were recorded at the times indicated by the letters on the graph below. The graphs below show variations of the CFP/YFP fluorescence emission ratio before, during and after IBMX application. **C**, comparison of the relative increase in CFP/YFP ratio in response to IBMX in control (n = 21) and AICAR-treated (n = 17) AVMs isolated from three different rats. Values are means±SEM. ***p<0.001 for differences from control values.

(p<0.05) and 39% (p<0.05) in both WT and KO mice respectively (Fig 7E). These results show that AMPKα2 activation is responsible for AC5 down-regulation in vivo in a pathological model of pressure overload while AC6 expression is not regulated by AMPKα2.

## Discussion

In this study we assessed the possible interaction between the cardiac beta-adrenergic and the AMPK pathways. The main results can be summarized as follows:

1) Using real time RT-PCR, we confirmed that AC5 expression is increased in the heart of mice deficient in the α2 subunit of AMPK. 2) The response of the slow calcium current $I_{Ca,L}$, to the PDE inhibitor IBMX, which reflects cAMP-dependent phosphorylation, was increased in AMPKα2-/- cardiomyocytes. 3) The cardiac contractile response to forskolin, an activator of AC, and to β-adrenergic stimulation was exacerbated in AMPKα2-/- mice compared to WT. 4) Conversely, stimulation of isolated adult cardiomyocytes with the AMPK activator

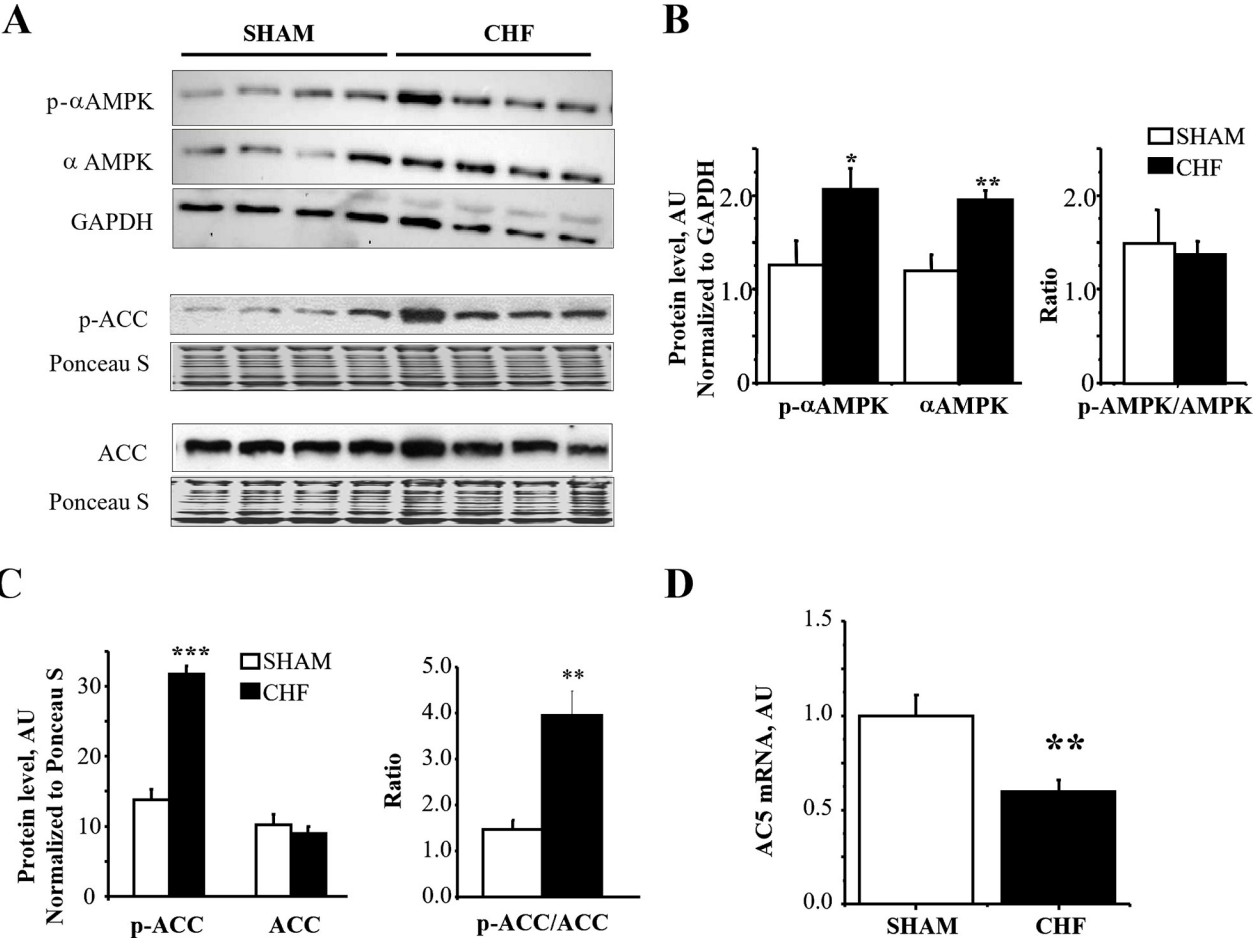

**Fig 6. AMPK phosphorylation and AC5 gene expression in hearts from sham and heart failure rats. A**, Representative western blots with phosphorylated-αAMPK (p-αAMPK), total αAMPK, phosphorylated-ACC (p-ACC), total ACC evidenced an increase in the phosphorylated forms of AMPK and ACC in HF rats; **B,** mean values of p-αAMPK, total αAMPK and p-αAMPK/αAMPK. **C,** mean values of p-ACC, total ACC and p-ACC/ACC. **D,** Adenylyl cyclase 5 (AC5) gene expression was measured by real-time RT-PCR and normalized to TBP. Values are means±SEM of 8 sham and 8 HF rats. *p<0.05, **p<0.01 and ***p<0.001 versus sham.

AICAR decreased the transcription of AC5 and blunted the cAMP response to IBMX. 5) In a rat model of systolic dysfunction, AC5 expression was decreased concomitant with an increase in AMPK phosphorylation. 6) Finally, the down-regulation of AC5 expression following TAC was blunted in AMPKα2-/- mice despite an exacerbated mechanical response to stress. Altogether these results show that the beta-adrenergic system is a target of AMPK in the heart. AMPK activation participate in the down-regulation of the beta-adrenergic pathway in the failing heart thus protecting it against energy wasting.

## AMPK, calcium current and contraction in KO mice

Global and constitutive deletion of the α2 subunit of AMPK does not induce overt cardiac defects in mice until the age of 17 weeks. Our mice thus display a milder cardiac phenotype than inducible cardiac-specific AMPKα2 knockout mice, which show impairment of cardiac contractile function and cardiac fibrosis two months after AMPKα2 deletion in the adulthood [8]. Our results are in accordance with previous studies showing that global and constitutive AMPKα2-/- mice do not exhibit anatomical or hemodynamic alteration nor decreased oxygen

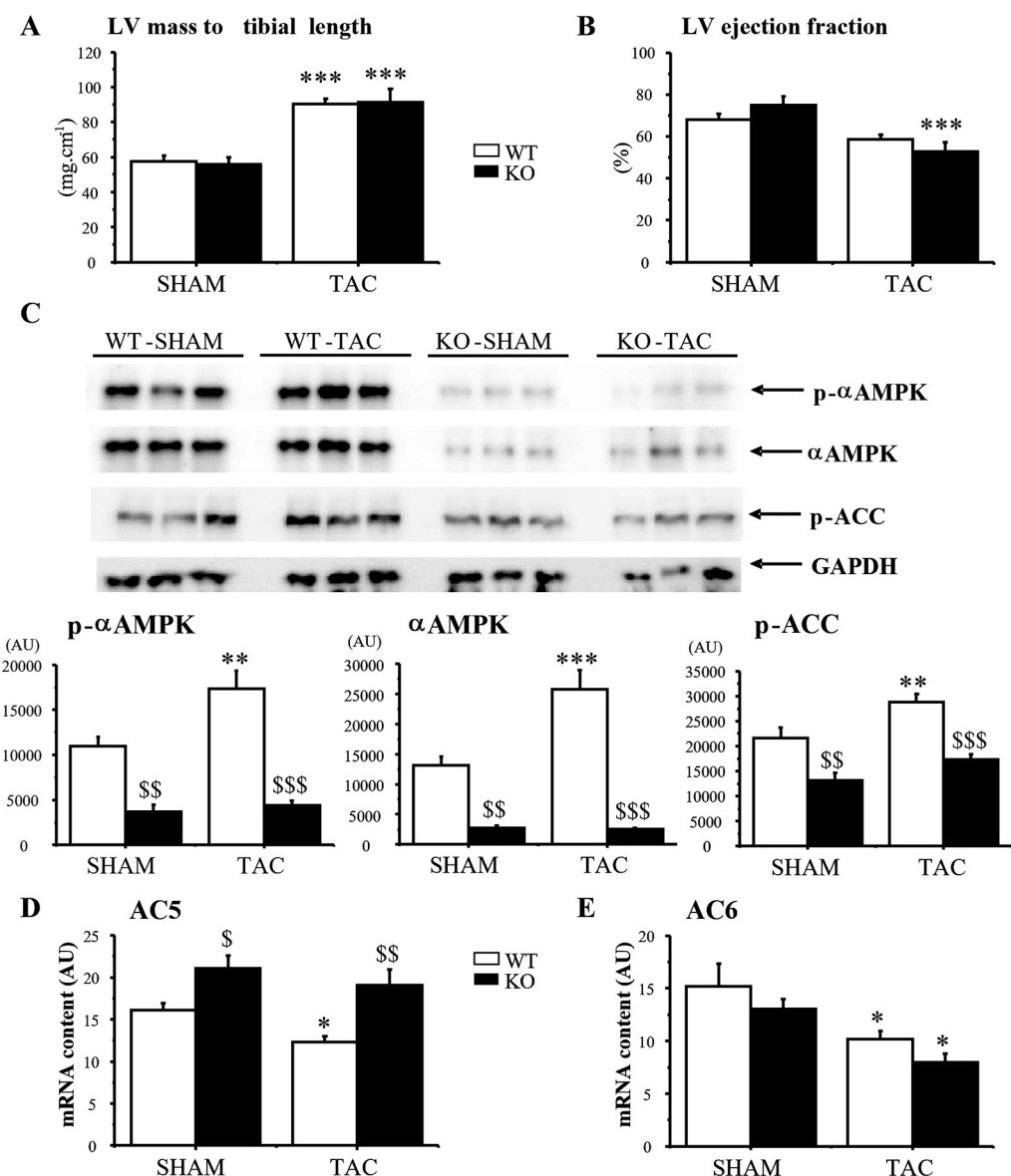

**Fig 7. AMPK phosphorylation and AC5 and AC6 gene expression in hearts from WT and AMPKα2-/- mice following TAC. A**, left ventricular mass/tibia length shows a similar hypertrophy in WT and KO mice; **B**, ejection fraction (%) showing the worsening of cardiac function in KO mice; **C**, representative western blots of αAMPK and ACC of three hearts per group performed on the same gel and mean values of phosphorylation evidencing an increase in αAMPK and ACC phosphorylation in WT-TAC mice only; **D**, following TAC, AC5 expression was significantly decreased in WT but not in KO mice while **E**, AC6 expression was decreased in both WT and KO mice. Values are means±SEM of 6–8 mice. *p<0.05, **p<0.01 and ***p<0.001 versus corresponding sham; $p<0.05, $$p<0.01 and $$$p<0.001 versus corresponding WT.

consumption or contractile dysfunction in isolated perfused hearts [16, 32]. Nonetheless they exhibit a decrease in maximal cardiac oxidative capacity linked to the down-regulation of cardiolipin biosynthesis rather than to decreased mitochondrial biogenesis [7] and, when submitted to stress, they show a diminished resistance to ischemia, and exacerbated pressure-overload-induced cardiac hypertrophy and contractile dysfunction [16, 32].

**Table 1. Echocardiographic measurements in WT and AMPKα2-/- mice following transverse aortic constriction.**

| | WT-SH | WT-TAC | KO-SH | KO-TAC |
|---|---|---|---|---|
| | N = 6 | N = 8 | N = 6 | N = 6 |
| **Heart rate (beats/min)** | 489±24 | 492±19 | **578±14**\* | **508±20**\* |
| **IVSd (mm)** | 0.70±0.05 | 0.83±0.06 | 0.76±0.02 | **0.98±0.08**\* |
| **IVSs (mm)** | 0.95±0.07 | 1.03±0.07 | 1.06±0.07 | 1.15±0.08 |
| **LVIDd (mm)** | 4.17±0.22 | 4.53±0.06 | 3.93±2.0 | 4.27±0.14 |
| **LVIDs (mm)** | 2.79±0.18 | 3.32±0.08 | 2.42±0.24 | **3.28±0.19**\*\* |
| **LVPWd (mm)** | 0.69±0.05 | **0.92±0.03**\*\* | 0.73±0.03 | **0.85±0.05**\* |
| **LVPWs (mm)** | 0.90±0.06 | 1.13±0.05 | **1.08±0.08**† | 1.09±0.06 |
| **LVTDV (ml)** | 0.19±0.03 | 0.23±0.01 | 0.16±0.02 | 0.20±0.02 |
| **LVTSV (ml)** | 0.06±0.01 | 0.09±0.01 | 0.04±0.01 | **0.09±0.02**\*\* |
| **EF (%)** | 67.8±3.0 | 58.7±2.2 | 74.8±4.1 | **52.7±4.5**\*\*\* |
| **Stroke volume (ml)** | 0.126±0.022 | 0.135±0.007 | 0.115±0.014 | **0.101±0.008**\* |
| **FS (%)** | 33.1±2.3 | 26.8±1.4 | 39.0±3.5 | **23.5±2.4**\*\*\* |
| **LV mass (mg)** | 103±6 | **164±13**\*\* | 102±9 | **157±16**\* |
| **CO (ml/min)** | 60.5±8.6 | 66.1±3.6 | 66.1±7.1 | 51.7±5.4 |

Data are expressed as means ± SEM. IVSd, interventricular septal thickness at end-diastole; IVSs, interventricular septal thickness at end-systole; LVIDd, left ventricular internal dimension at end-diastole; LVIDs, left ventricular internal dimension at end-systole; LVPWd, left ventricular posterior wall thickness at end-diastole; LVPWs, left ventricular posterior wall thickness at end-systole; LVTDV, left ventricular telediastolic volume, LVTSV, left ventricular telesystolic volume; EF, ejection fraction; FS, fractional shortening; CO, cardiac output. N = number of animals.

\*P<0.05

\*\*P<0.01

\*\*\*P<0.001 vs sham

†P<0.05 vs WT.

For the first time in AMPKα2 deficient mice, we identified an increase in AC5 but not AC6 expression in the heart. AC5 and AC6 are the main AC isoforms expressed in the cardiomyocyte and constitute a cardiac subgroup of ACs that are stimulated by NaF, GTPrS, and forskolin, and inhibited by adenosine or calcium [33]. AC5 possess a much higher catalytic activity than AC6, is more cardiac-specific and highly expressed in adults [21]. Experiments with genetically modified animal models have clearly evidenced distinct functions for each AC isoform. Indeed, targeted deletion of AC5 is associated with cardiac protection in response to pressure overload and stress like long-term β-adrenergic stimulation [34] while its overexpression sensitizes to chronic β-adrenergic stimulation [35] and promotes cardiac arrhythmogenesis [36]. By contrast, deletion in AC6 is accompanied by reduced left ventricular contraction and relaxation [37]. In our study, the fact that, in AMPKα2 KO-TAC mice, only the decrease in AC5 is cancelled but not the decrease in AC6, confirms that the specificity of the AMPKα2-AC5 axis of regulation is maintained in pathological state [38].

We next investigated whether AMPK-induced alteration of AC gene expression could be of physiological significance. For this we measured the response of $I_{CaL}$, one of the main targets of AC-dependent cAMP increase. In basal state, i.e. without β-adrenergic stimulation we did not observe a change in the calcium current amplitude in KO versus WT mice, in accordance with the fact that the basal cyclase activity is quite low in adult ventricular cells and is totally counterbalanced by PDE activity. However, inhibiting phosphodiesterases that degrade cAMP revealed a basal turnover of the cyclase, as the calcium current was significantly increased. The greater increase of $I_{Ca,L}$ in response to IBMX in KO mice shows that the turnover of cAMP is higher when AMPKα2 is deleted, in line with the observed increase in AC5 expression. This

shows a new role of AMPK in cAMP signaling in cardiac cells that echoes the activation of phosphodiesterase 4B (PDE4B) by AMPK described in the hepatocytes [39].

## AMPK activation decreases AC5 expression and cAMP content

While AMPKα2 deletion induces an upregulation of AC5 and increased β-adrenergic activation, it was important to know whether activation of AMPK would be able to down-regulate AC5 and to blunt β-adrenergic activation. We show that this is indeed the case as incubation of rat AVMs with the pharmacological activator of AMPK, AICAR, induced a decrease in AC5 expression. By extrapolation, knowing that it has been suggested that the loss of AC5 protects against diabetic cardiomyopathy [40], this effect of AMPK on AC5 expression could participate in the cardioprotective effect of metformin, an AMPK activator, in diabetes models [41].

In order to know whether AC5 expression changes influence cAMP content, we infected rat AVMs with an adenovirus expressing the cAMP probe Epac2-camps to monitor the intracellular content of cAMP by FRET [42]. Stimulation of cardiomyocytes with AICAR blunted the IBMX-induced cAMP increase compared to untreated cells. This clearly evidenced that AC5 expression and function can be positively and negatively modulated by AMPK. Interestingly, it was shown that cAMP can negatively affect AMPK activity by phosphorylation of AMPKα2 Ser$^{491}$ [43]. Here we show that in turn, AMPK can modulate the β-adrenergic pathway by controlling the expression of AC5. This reveals a new type of interplay between these two pathways.

## AMPK, cardiac dysfunction and pressure overload

It is well recognized that the failing myocardium is energy starved. Decreased energy production, increased energy consumption and energy wastage characterize the failing heart [23, 44]. This energy starvation state leads to decreased ATP/AMP ratio [45] and activation of AMPK [13] as a compensatory mechanism. We show that AMPK is activated in pressure overload-induced myocardial dysfunction as the phosphorylated form of AMPK is increased together with the content of α subunits and the phosphorylation of its down-stream target ACC. Increased total AMPKα has also been reported in mice following TAC [16], and following pressure overload in rats [13]. It thus appears that AMPK activation in chronic heart failure is caused by an increase in total AMPK content, rather than by a selective increase in the phosphorylated form. It can be speculated that while activation of AMPK by post-translational modifications could be an acute response to energy deficiency or increase in workload, in the long term as in hypertrophy or heart failure, energy deficiency induces transcriptional up-regulation of AMPK subunits leading to an increase in the active form of the enzyme despite maintained phosphorylation ratio.

As previously shown [16], in AMPKα2-/- mice the TAC-induced decrease in left ventricular function was exacerbated (ejection fraction decreased by 13% (ns) in WT mice compared to 30% (p<0.001) in KO mice) suggesting a protective role of AMPK. Such effect of AMPK activation in HF has already been suggested. For example, the lowering of the AMPK signaling in adiponectin-deficient mice is responsible for progressive deterioration of HF [46]. AMPKα2 deficiency exacerbates pressure-overload-induced left ventricular hypertrophy and dysfunction in mice [16]. Conversely, metformin, an AMPK activator, attenuates oxidative stress-induced cardiomyocyte apoptosis, improves left ventricular function and prevents progression of HF in dogs [14, 15]. In addition, activation of AMPK by AICAR attenuates progression of pressure overload-induced HF [47] and AMPK could be at the heart of the beneficial effects of specific pharmaceutical compounds used to treat cardiac diseases [48, 49]. All these results support the contention that energy deficiency and unbalance is an important

pathological factor in HF and that activation of AMPK is a protective mechanism and should be considered as a key therapeutic target.

## AC5, AMPK, cardiac dysfunction and pressure overload

It is well known that the failing heart suffers from a β-adrenergic overdrive. In response to chronic stimulation, cells have developed strategies to decrease the intracellular signaling pathways and minimize the adrenergic signals. This includes desensitization, internalization and down-regulation of the β-receptors, increased inhibitory and decreased stimulatory G-proteins and G-protein coupled receptor kinases (for reviews see [19, 22]). In pacing-induced HF, a decrease in forskolin or G-protein-stimulated AC activity was evidenced long time ago [50, 51] associated with AC5 downregulation [18]. Deletion of AC5 protects the heart from stress and contractile dysfunction [52] showing that AC5 down-regulation is beneficial in HF. Interestingly, it was shown that disruption of AC5 results in more effective desensitization of the cAMP signal and protection against cardiomyocyte apoptosis after long-term catecholamine stress [52].

We observed a down-regulation of AC5 expression in the mouse pressure overload and rat myocardial dysfunction models, concomitant with the activation of AMPK. Interestingly, AC5 expression was down-regulated in WT but not in AMPKα2-/- mice strongly suggesting a mechanistic link between AMPK activation and AC5 downregulation in pressure overload-induced myocardial dysfunction. Although the beneficial effects of AMPK activation in HF can be linked to its capacity to increase energy production, it is also known that AMPK exerts energy sparing effects by reducing energy consuming phenomena. Regulation of AC5 by AMPK could be such a mechanism participating in desensitization of the β-adrenergic pathway in HF.

It was shown that in failing hearts, the increased cardiac work induced by β-adrenergic stimulation increases the energy cost of contraction [53]. The energy wasting effect of excess β-adrenergic stimulation is due to accelerated heart rate, increased energy cost of cellular calcium homeostasis and contraction, deteriorated diastolic filling and decreased blood flow [22]. As β-adrenergic stimulation is highly energy consuming, blunting the β-adrenergic system could be a strategy to preserve the compromised energetic balance in HF. Desensitization of the β-adrenergic-AC-cAMP pathway protects cardiomyocytes from arrhythmogenic, energy-consuming, hypertrophic and apoptotic effects of excess catecholamines [22]. Because AC5 inhibitors could exert similar benefits as beta-blockers, potentially with less negative inotropic effects during HF, it was thus proposed that ACs could be considered as new drug targets [19, 54]. It is thus tempting to speculate that down-regulation of AC5 when AMPK is activated represent a new energy sparing effect of AMPK. Different mechanisms have been proposed for the beneficial effects of AMPK in HF including increased glucose metabolism, activation of NOS and PGC-1alpha, anti-apoptotic effects, limitation of hypertrophic growth, increased mitochondrial respiration and mitochondrial protein expression [15, 55]. In addition to these mechanisms, this study shows that blunting of the beta-adrenergic pathway could represent an additional beneficial effect of AMPK activation in HF. Moreover, the increase in the expression of genes involved in mitochondrial biogenesis and function in skeletal muscle of AC5KO mice suggests that AC5 could directly or indirectly impacts energy capacity production of the cell and its down-regulation could be important in the stimulation of energy production processes by AMPK [56].

Interestingly, the antidiabetic drugs metformin and AICAR, two AMPK activators, reduce apoptosis and prevent HF in experimental models, also suggesting a protective role of AMPK activation. Metformin was thus proposed as a new therapy for HF [14, 15]. Part of these effects

could be mediated by the β-adrenergic desensitizing effects of AMPK activation. Similarly, the observed cardioprotective effect of AMPK against pressure-overload-induced ventricular hypertrophy and dysfunction [16] can also be attributed in part to the AMPK-induced β-adrenergic pathway down-regulation.

In summary, these results show that AMPK activation leads to a down-regulation of AC5 in the heart and a blunted response to β-adrenergic stimulation. This uncovers a new mechanism linking energy starvation with β-adrenergic desensitization in pressure overload and cardiac dysfunction. This highlights a key crosstalk between AMPK and AC5 the mechanistic links of which remain to be investigated in further studies.

## Supporting information

**S1 Raw images.**
(PDF)

## Acknowledgments

We are indebted to Dominique Fortin, Florence Lefebvre, Jocelyne Leclerc and M-F Messmer for helpful technical assistance. We thank Valérie Domergue and the animal core facility of UMR IPSIT for efficient handling and breeding of the animals. We are grateful to Pauline Robert for technical assistance in mice genotyping, and Patrick Lechêne for FRET experiments. The authors would also like to thank R. Fischmeister and Ana-Maria Gomez for continuous support and fruitful discussions. Our laboratory is a member of the laboratory of LERMIT. R V-C is emerit scientist at Centre National de la Recherche Scientifique (CNRS).

## Author Contributions

**Conceptualization:** Anne Garnier, Jérôme Leroy, Benoit Viollet, Vladimir Veksler, Mathias Mericskay, Renée Ventura-Clapier, Jérôme Piquereau.

**Data curation:** Anne Garnier, Jérôme Leroy, Claudine Deloménie, Philippe Mateo.

**Formal analysis:** Anne Garnier, Jérôme Leroy, Philippe Mateo, Jérôme Piquereau.

**Funding acquisition:** Anne Garnier, Mathias Mericskay, Renée Ventura-Clapier.

**Investigation:** Anne Garnier, Jérôme Leroy, Jérôme Piquereau.

**Methodology:** Anne Garnier, Jérôme Leroy, Renée Ventura-Clapier.

**Resources:** Anne Garnier.

**Supervision:** Anne Garnier, Mathias Mericskay, Renée Ventura-Clapier, Jérôme Piquereau.

**Validation:** Anne Garnier, Jérôme Leroy, Claudine Deloménie, Philippe Mateo, Vladimir Veksler, Renée Ventura-Clapier, Jérôme Piquereau.

**Visualization:** Renée Ventura-Clapier, Jérôme Piquereau.

**Writing – original draft:** Anne Garnier, Jérôme Leroy, Benoit Viollet, Mathias Mericskay, Renée Ventura-Clapier, Jérôme Piquereau.

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
