## [Decision Letter · Decision Letter 0]

18 Jul 2023

PONE-D-23-18218MODULATION OF CARDIAC cAMP SIGNALING BY AMPK AND ITS ADJUSTMENTS IN PRESSURE OVERLOAD-INDUCED MYOCARDIAL DYSFUNCTION IN RAT AND MOUSEPLOS ONE

Dear Dr. Piquereau,

Thank you for submitting your manuscript to PLOS ONE. After careful consideration, we feel that it has merit but does not fully meet PLOS ONE’s publication criteria as it currently stands. Therefore, we invite you to submit a revised version of the manuscript that addresses the points raised during the review process.

We look forward to receiving your revised manuscript.

Kind regards,

Luis Eduardo M Quintas, Ph.D.

Academic Editor

PLOS ONE

“We are indebted to Dominique Fortin, Florence Lefebvre, Jocelyne Leclerc and M-F Messmer for helpful technical assistance. We thank Valérie Domergue and the animal core facility of UMR IPSIT for efficient handling and breeding of the animals. We are grateful to Pauline Robert for technical assistance in mice genotyping, and Patrick Lechêne for FRET experiments. The authors would also like to thank R. Fischmeister and Ana-Maria Gomez for continuous support and fruitful discussions.

              Our laboratory is a member of the Laboratory of LERMIT and is supported by grants from “Fondation pour la Recherche Médicale” (to A.G, #DPM20121125546), European Research Area Networkon Cardiovascular Diseases (to J.P, #ANR-19-ECVD-0007-01), Fondation de France (to A.G, 00086500). R V-C is emerit scientist at Centre National de la Recherche Scientifique (CNRS).”

Please include your amended statements within your cover letter; we will change the online submission form on your behalf."

Reviewers' comments:

Reviewer's Responses to Questions

**Comments to the Author**

1. Is the manuscript technically sound, and do the data support the conclusions?

Reviewer #1: Yes

2. Has the statistical analysis been performed appropriately and rigorously? 

Reviewer #1: Yes

3. Have the authors made all data underlying the findings in their manuscript fully available?

Reviewer #1: Yes

4. Is the manuscript presented in an intelligible fashion and written in standard English?

Reviewer #1: Yes

5. Review Comments to the Author

Reviewer #1: In this article, the authors worked with the hypothesis of a possible crosstalking between AC5 and AMPK in hearts of WT and KO mice for AMPKα2-/-. The work was very well executed, it is very well written and presents very robust data to demonstrate the initial hypothesis. The authors worked very well on the intracellular signaling pathways involved in this crosstalking, using various resources to reach their conclusions.

Minor points to consider.

The Ca2+ current explored by the authors as a way to demonstrate the importance of the action of AMPK on AC5 showed no difference between the WT and KO animals initially tested (Fig.2C). The difference appeared only in KO animals when IBMX (PDE inhibitor) was used (Fig.2D). Shouldn't the increased Ca2+ current in the KO animals have appeared in both moments? How in KO animals would PDE be masking this situation? The authors should explore this situation further since this finding was not discussed.

6. PLOS authors have the option to publish the peer review history of their article (what does this mean?). If published, this will include your full peer review and any attached files.

Reviewer #1: No

---

## [Author Response · Author response to Decision Letter 0]

27 Aug 2023

Reviewer #1: In this article, the authors worked with the hypothesis of a possible crosstalking between AC5 and AMPK in hearts of WT and KO mice for AMPKα2-/-. The work was very well executed, it is very well written and presents very robust data to demonstrate the initial hypothesis. The authors worked very well on the intracellular signaling pathways involved in this crosstalking, using various resources to reach their conclusions.

Minor points to consider.

The Ca2+ current explored by the authors as a way to demonstrate the importance of the action of AMPK on AC5 showed no difference between the WT and KO animals initially tested (Fig.2C). The difference appeared only in KO animals when IBMX (PDE inhibitor) was used (Fig.2D). Shouldn't the increased Ca2+ current in the KO animals have appeared in both moments? How in KO animals would PDE be masking this situation? The authors should explore this situation further since this finding was not discussed.

Response : We thank the reviewer for this relevant comment. Actually, there is no effect on basal current because, unlike in atrial cells, basal cyclase activity is quite low in adult ventricular cells and is totally counterbalanced by PDE activity. Although cyclase activity is higher in KOs, it is unlikely to be sufficient to surpass PDE activity. It is therefore logical to see an increase in cAMP levels measured by FRET, and consequently increased calcium channel activity, only when PDEs are inhibited. This more detailed explanation has been included in the text p. 18

---

## [Editor Report · Decision Letter 1]

11 Sep 2023

Modulation of cardiac cAMP signaling by AMPK and its adjustments in pressure overload-induced myocardial dysfunction in rat and mouse

PONE-D-23-18218R1

Dear Dr. Piquereau,

We’re pleased to inform you that your manuscript has been judged scientifically suitable for publication and will be formally accepted for publication once it meets all outstanding technical requirements.

Kind regards,

Luis Eduardo M Quintas, Ph.D.

Academic Editor

PLOS ONE
---

## [Editor Report · Acceptance letter]

13 Sep 2023

PONE-D-23-18218R1 

Modulation of cardiac cAMP signaling by AMPK and its adjustments in pressure overload-induced myocardial dysfunction in rat and mouse 

Dear Dr. Piquereau:

I'm pleased to inform you that your manuscript has been deemed suitable for publication in PLOS ONE. Congratulations! Your manuscript is now with our production department. 

Kind regards, 

on behalf of

Dr. Luis Eduardo M Quintas 

Academic Editor

PLOS ONE